# A simple and economic protocol for efficient *in vitro* fertilization using cryopreserved mouse sperm

Magdalena Wigger[1,2], Simon E. Tröder●[1,2‡]*, Branko Zevnik●[1,2‡]*

**1** Cluster of Excellence Cellular Stress Responses in Aging-Associated Diseases (CECAD), Faculty of Medicine and University Hospital Cologne, University of Cologne, Cologne, Germany, **2** in vivo Research Facility, Faculty of Medicine and University Hospital Cologne, University of Cologne, Cologne, Germany

‡ These authors are joint senior authors on this work.
* branko.zevnik@uk-koeln.de (BZ); simon.troeder@uk-koeln.de (SET)

## Abstract

The advent of genome editing tools like CRISPR/Cas has substantially increased the number of genetically engineered mouse models in recent years. In support of refinement and reduction, sperm cryopreservation is advantageous compared to embryo freezing for archiving and distribution of such mouse models. The *in vitro* fertilization using cryopreserved sperm from the most widely used C57BL/6 strain has become highly efficient in recent years due to several improvements of the procedure. However, purchase of the necessary media for routine application of the current protocol poses a constant burden on budgetary constraints. In-house media preparation, instead, is complex and requires quality control of each batch. Here, we describe a cost-effective and easily adaptable approach for *in vitro* fertilization using cryopreserved C57BL/6 sperm. This is mainly achieved by modification of an affordable commercial fertilization medium and a step-by-step description of all other necessary reagents. Large-scale comparison of fertilization rates from independent lines of genetically engineered C57BL/6 mice upon cryopreservation and *in vitro* fertilization with our approach demonstrated equal or significantly superior fertilization rates to current protocols. Our novel SEcuRe (Simple Economical set-up for Rederivation) method provides an affordable, easily adaptable and harmonized protocol for highly efficient rederivation using cryopreserved C57BL/6 sperm for a broad application of colony management in the sense of the 3Rs.

**Data Availability Statement:** All relevant data are within the manuscript and its Supporting Information files.

**Funding:** The research was supported by the Deutsche Forschungsgemeinschaft (DFG, German

## Introduction

Over the past decades genetically engineered mouse models (GEMMs) have grown to be the key experimental tools in biomedical research. The rise of highly versatile genome-editing technologies in the recent years has led to a rapid expansion of the number of mutant mouse lines available [1]. To meet the demand for efficient and cost-effective archiving, distribution and colony management of these strains, cryopreservation of sperm for subsequent rederivation via *in vitro* fertilization (IVF) has become the method of choice. In contrast to embryos

Research Foundation, https://www.dfg.de) under (1) KFO 329 to BZ, (2) Germany´s Excellence Strategy – EXC 229 – Cellular Stress Responses in Aging Associated Diseases, Research Area F01 to BZ, (3) Germany´s Excellence Strategy – EXC 2030/1 – 390661388 to BZ. The funders had and will not have a role in study design, data collection and analysis, decision to publish, or preparation of the manuscript.

**Competing interests:** The authors have declared that no competing interests exist.

the cryopreservation of sperm does not involve breeding efforts, requires far less animals and enables generation of a large number of embryos upon rederivation [2].

The first successful cryopreservation of mouse sperm using a cryoprotective agent (CPA) composed of 18% raffinose and 3% skim milk, adopted by virtually all IVF laboratories, was published in 1990 [3]. However, for a long time frozen-thawed sperm of inbred C57BL/6 mice, a genetic background most widely used for GEMMs by researchers and large consortia like EUCOMM and KOMP, have suffered from strain specific low fertilization rates after thawing [4–6]. During the last two decades many improvements to sperm cryopreservation and preincubation, as well as to the IVF procedure were introduced which led to a dramatic increase in the number of oocytes fertilized by sperm originating from problematic strains, including C57BL/6.

For sperm cryopreservation, CPA composed of raffinose and skim milk has been supplemented with either monothioglycerol (MTG) [7] or L-glutamine [8]. These compounds were able to strongly enhance the protective effect of the original CPA against cellular injury which is believed to be mediated by the antioxidative ability of MTG or a membrane stabilizing function of L-glutamine resulting in elevated fertilization rates [7, 8].

Another advancement concerned the sperm preincubation in order to allow the capacitation process, a prerequisite for fertilization, which under physiological conditions takes place in the female reproductive tract [9, 10]. One major step of capacitation is considered to involve cholesterol removal from the sperm plasma membrane. *In vitro* this process can be recapitulated in TYH medium [11] containing bovine serum albumin (BSA) which acts as a cholesterol acceptor [12]. Since BSA can be contaminated by other serum components [13], a protein-free version of TYH medium, termed c-TYH has been established by replacing BSA with polyvinyl alcohol (PVA) and an alternative cholesterol acceptor–methyl-β-cyclodextrin (MBCD) [14]. Pre-treatment of frozen-thawed mouse sperm in this medium has led to an improvement of capacitation and as a consequence to an increase of the fertilization rate *in vitro* [15].

The composition of the IVF medium used in conjunction with frozen-thawed sperm has been under investigation as well. Most commonly, HTF medium based on the chemical composition of human tubal fluid [16] is used in mouse IVF [17, 18]. Elevation of the calcium concentration from default 2.04 mM to 5.14 mM has been shown to enhance the fertilization rate of a variety of inbred strains including C57BL/6J [19]. A further advancement arose with the supplementation of the fertilization medium with reduced glutathione (GSH) [20]–an antioxidant–following the hypothesis that frozen-thawed mouse sperm similarly to sperm of other species undergo oxidative stress [21, 22]. In fact, GSH reduced the redox imbalance and increased fertilization capacity of cryopreserved sperm [20]. In addition, GSH may be facilitating sperm penetration by reducing disulfide bonds in the zona pellucida to promote fertilization [23].

Two protocols for sperm cryopreservation and IVF incorporating these improvements are commonly used in the field [17]. The first one developed by Ostermeier *et al.* at The Jackson Laboratory (JAX) is based on the addition of MTG to the CPA [7, 24, 25]. It has a broad applicability as it employs only one type of medium (i.e., HTF) for sperm preincubation and fertilization. This medium can be substituted by commercially available Research Vitro Fert (RVF; Cook Medical) medium further supporting the simplicity of the method. The second protocol established by the Nakagata laboratory at the Center for Animal Resources and Development (CARD) is more complex as it uses CPA supplemented with L-glutamine (gCPA), c-TYH medium for capacitation and HTF with an elevated calcium concentration and GSH supplementation for fertilization [8, 23, 26, 27]. The latter method, referred to as the CARD protocol, has been predominantly adapted by many repositories such as the European Mouse Mutant Archive (EMMA). For sporadic application all components are commercially available as a

ready-to-use set (so-called "FERTIUP® CPA" and "FERTIUP® Preincubation Medium (PM)–CARD MEDIUM® Set"). Alternatively, for laboratories routinely performing IVF a recent protocol describes the preparation of all necessary reagents in a cost-effective fashion [27]. However, the composition of complex and high-quality fertilization media can be challenging and requires quality control of each batch prior to its use [17].

Therefore, we were seeking to establish a simplified and cost-efficient method for IVF using cryopreserved C57BL/6 sperm for routine application based on the CARD protocol. As a consequence, we developed a Simple Economical set-up for Rederivation (SEcuRe) approach with an inexpensive commercially available fertilization medium (i.e., RVF medium) recommended and used by many laboratories including JAX [25] and modified by us according to the CARD method (i.e., with an elevated calcium concentration and GSH supplementation) which we termed modified RVF (mRVF). Results presented here are based on the retrospective study of routine cryopreservation and IVF with different protocols performed in our laboratory over several years employing a substantial number of GEMMs. The robust data of this large-scale comparison is therefore likely to reflect the daily routine in other mouse IVF laboratories. We validated our method by demonstrating equal fertilization rates *in vitro* and birth rates *in vivo* to those obtained when sperm cryopreservation and IVF were performed with the FERTIUP® PM–CARD MEDIUM® Set (referred to as the CARD Set protocol). Additionally, we conducted a side-by-side analysis of FERTIUP® PM and in-house prepared c-TYH. This way we provided researchers with a less complicated (when FERTIUP® PM is utilized) and more economical (c-TYH) approach to IVF recoveries. The protocol includes the convenient preparation of c-TYH medium from frozen stocks which we adapted analogous to popular protocols for the in-house production of embryo culture media like KSOM [17]. Moreover, the present study demonstrates to our knowledge the first direct comparison of fertilization efficiencies of the two most commonly used methods for sperm cryopreservation and IVF using frozen-thawed sperm (i.e., the CARD Set and the Ostermeier *et al*. protocol) and points to an advantage of the CARD-based protocols like our SEcuRe approach for C57BL/6 mice (Fig 1).

## Materials and methods

The protocol described in this peer-reviewed article is published on protocols.io, dx.doi.org/10.17504/protocols.io.bx2spqee and is included for printing as S1 File with this article.

### Ethical statement

All animal protocols were performed in compliance with the European, national and institutional guidelines and approved by the State Office of North Rhine-Westphalia, Department of Nature, Environment and Consumer Protection (LANUV NRW, Germany; animal study protocol AZ 84–02.04.2014.A372 and AZ 81–02.04.2019.A335). Mice were euthanized by cervical dislocation. All efforts were made to minimize suffering of animals used. Animals were maintained in the CECAD Research Center, University of Cologne, Germany, in individually ventilated cages (Greenline GM500; Tecniplast) at 22°C (± 2°C) and a relative humidity of 55% (± 5%) under 12-hour light cycle on sterilized bedding (Aspen wood, Abedd, Germany) and with access to sterilized commercial pelleted diet (Ssniff Spezialdiäten GmbH) and acidified water *ad libitum*. The microbiological status was examined as recommended by Federation of European Laboratory Animal Science Associations (FELASA) and the mice were free of all listed agents including opportunists [28]. Ketamine (100 mg/kg body weight; Ketaset, Zoetis Deutschland GmbH) and xylazine chloride (10 mg/kg BW; XYLAZIN, Serumwerk Bernburg AG) were used as anesthetics and carprofen (5 mg/kg BW; Carprosol, CP-Pharma Handels-Gesellschaft mBH) as analgesic after surgery.

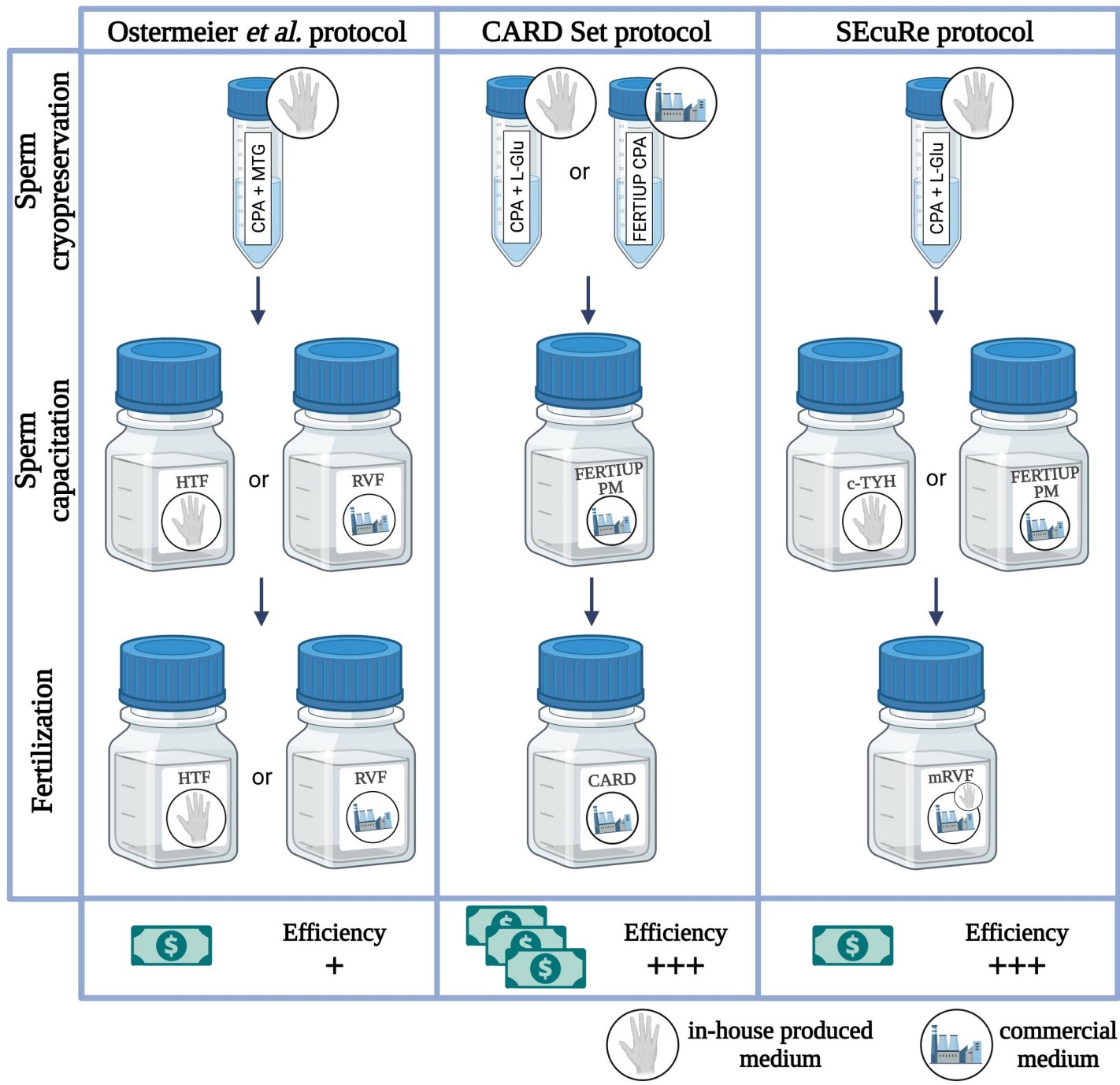

**Fig 1. Comparison of the Ostermeier *et al.*, CARD Set and SEcuRe protocols.** CPA = cryoprotective agent; MTG = monothioglycerol; L-Glu = L-glutamine; RVF medium = Research Vitro Fert fertilization medium; mRVF medium = modified RVF fertilization medium with elevated calcium concentration (5.14 mM) and GSH supplementation.

## Media preparation

For the CARD Set protocol the FERTIUP® PM—CARD MEDIUM® Set (i.e., FERTIUP® PM as mouse sperm preincubation medium and CARD MEDIUM® as fertilization medium; Cosmo Bio; KYD-004-EX) was used according to the manufacturer's instructions. L-glutamine

containing CPA (gCPA) was prepared as described below for the SEcuRe approach. An explanation of media preparation according to the Ostermeier *et al.* protocol has previously been published [24, 25].

The following description details the SEcuRe protocol developed in this study.

**Sperm cryopreservation medium.** For the cryopreservation of sperm gCPA containing 18% raffinose pentahydrate and 3% skim milk supplemented with 100 mM L-glutamine was prepared as published [18, 27]. Briefly, 0.146 g of L-glutamine (Sigma; G8540) was placed in 10 ml of prewarmed (60°C) water (Sigma; W1503) and vortexed for 3 min. Subsequently, 1.8 g of raffinose pentahydrate (Sigma; R7630) and 0.3 g of skim milk (Becton Dickinson; 232100) was added, the solution vortexed for 3 min and incubated for 90 min at 60°C. gCPA was vortexed every 30 min for 3 min. Next, the solution was centrifuged at 10,000 g for 60 min and the supernatant was filtered through a 0.22 μm filter (PALL; 4652). After osmolality check (500–520 mOsm/kg) aliquots were stored at room temperature for up to 3 months.

**Sperm capacitation medium.** FERTIUP® PM (Cosmo Bio; KYD-002-EX-X5) or c-TYH containing 1.0 mg/ml of polyvinyl alcohol (PVA; Sigma; P8136) and 0.75 mM methyl-β-cyclodextrin (MBCD; Sigma; C4555) was used for sperm capacitation after thawing. The composition of c-TYH medium was as published previously [14] but prepared from concentrated stocks as described in details in the Supporting Information section (S1 File) and on protocols. io, dx.doi.org/10.17504/protocols.io.bx2spqee.

**Fertilization medium.** For the IVF procedure a commercial RVF fertilization medium (Research Vitro Fert, Cook Medical; K-RVFE-50) was supplemented with 1 mM (frozen sperm) or 0.25 mM (freshly harvested sperm) reduced GSH (Sigma; G4251) and the calcium concentration was increased from default 2.04 mM to 5.14 mM (mRVF). Therefore, a 100x $CaCl_2$ stock solution (310 mM) was prepared by dissolving 0.4558 g of $CaCl_2$ (Sigma; C7902) in 10 ml of water (Sigma; W1503). The solution was filtered through a 0.22 μm filter and aliquots were stored at -20°C for a maximum of 6 months. On the day of IVF an aliquot of $CaCl_2$ was thawed at room temperature. Subsequently, 150 μl of 100x $CaCl_2$ was added to 15 ml of fertilization medium and mixed gently. Next, 1 ml of fertilization medium supplemented with $CaCl_2$ was placed in a tube containing 30.7 mg of GSH and vortexed. 50 μl (frozen sperm) or 10 μl (freshly harvested sperm) of this solution was added to 5 ml (frozen sperm) or 4 ml (freshly harvested sperm) of fertilization medium supplemented with $CaCl_2$, mixed gently and filtered using 0.22 μm syringe end filter.

## Sperm freezing/Thawing and capacitation

For the CARD Set protocol sperm cryopreservation and preincubation were performed according to the manufacturer's instructions (Cosmo Bio; KYD-004-EX) as well as described [27]. A step-by-step explanation of the sperm freezing/thawing and capacitation protocol employed in the Ostermeier *et al.* method has been described previously [24, 25].

The following description details the SEcuRe protocol developed in this study (Fig 2A and 2B). Freezing straws (Minitüb GmbH; 13407/0010) were prepared as following. 20 straws for 2 sacrificed males were marked at 2.3 cm and 4.0 cm at the open end and labeled at the other end (cotton plug). A 1 ml syringe was attached to the labeled end of the straw and RVF medium was aspirated until the meniscus reached the 4.0 cm mark. Then 2.3 cm air was drawn into the straw and the assembly was stored until required. Subsequently, 120 μl drop of gCPA was placed in a 35-mm culture dish and covered with paraffin oil (Sigma; 76235). In order to obtain a tall, semi-spherical drop another 120 μl of gCPA was added to reach a final volume of 240 μl (for 4 cauda epididymides pooled from 2 males of the same genotype).

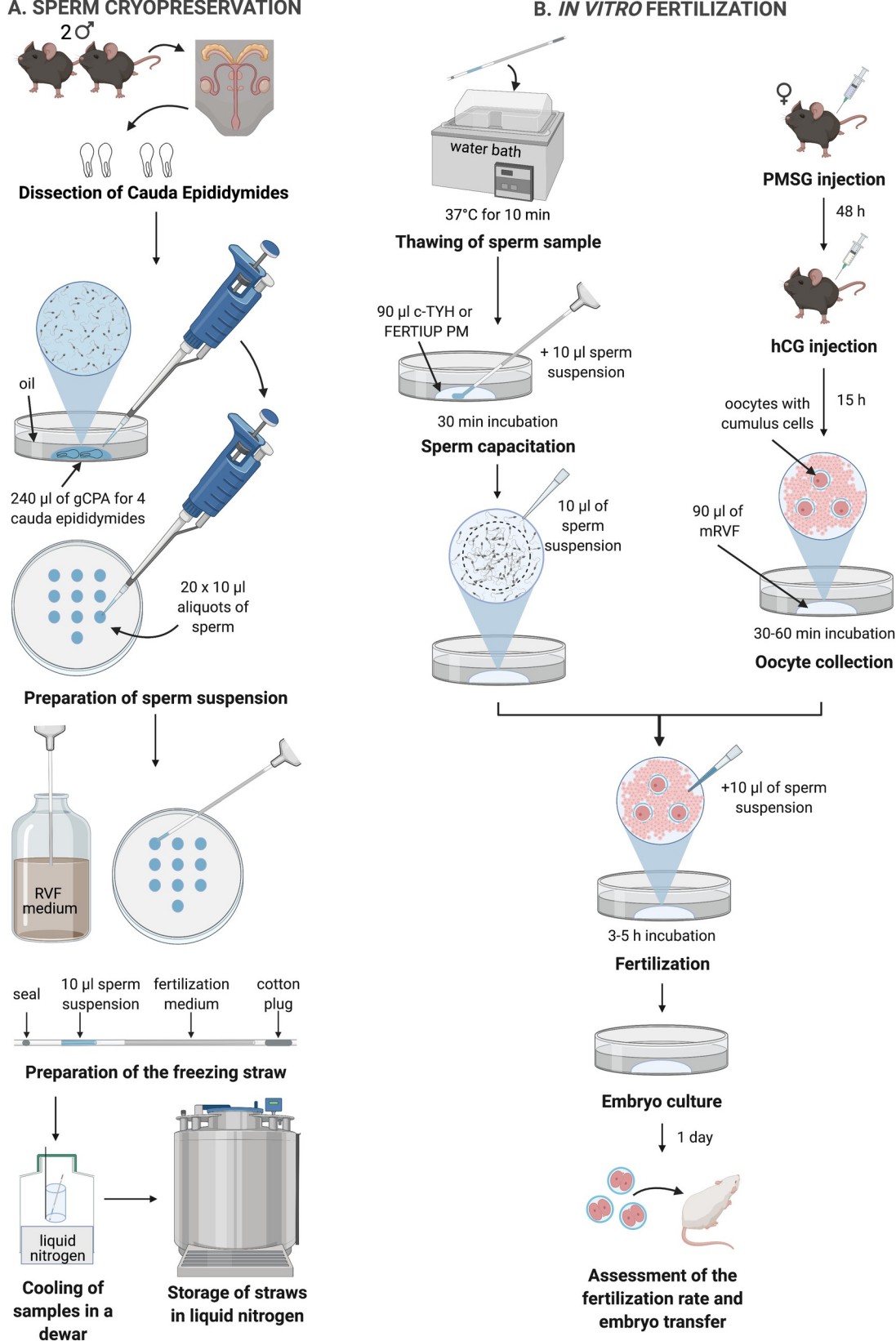

**A. SPERM CRYOPRESERVATION**

2♂

Dissection of Cauda Epididymides

oil

240 µl of gCPA for 4 cauda epididymides

20 x 10 µl aliquots of sperm

Preparation of sperm suspension

RVF medium

seal | 10 µl sperm suspension | fertilization medium | cotton plug

Preparation of the freezing straw

liquid nitrogen

Cooling of samples in a dewar

Storage of straws in liquid nitrogen

**B. *IN VITRO* FERTILIZATION**

water bath

37°C for 10 min

Thawing of sperm sample

90 µl c-TYH or FERTIUP PM

+ 10 µl sperm suspension

30 min incubation

Sperm capacitation

10 µl of sperm suspension

PMSG injection

48 h

hCG injection

oocytes with cumulus cells

15 h

90 µl of mRVF

30-60 min incubation

Oocyte collection

+10 µl of sperm suspension

3-5 h incubation

Fertilization

Embryo culture

1 day

Assessment of the fertilization rate and embryo transfer

**Fig 2. Scheme of the SEcuRe protocol.** (A) Sperm cryopreservation, (B) *in vitro* fertilization using frozen-thawed sperm. gCPA = cryoprotective agent supplemented with L-glutamine; RVF medium = Research Vitro Fert fertilization medium; mRVF = modified RVF fertilization medium with elevated calcium concentration (5.14 mM) and GSH supplementation.

For each cryopreservation two 10- to 20-week-old C57BL/6 mutant males of the same line (or wild-type C57BL/6NRj males in case of FERTIUP® PM and c-TYH side-by-side comparison) were sacrificed by cervical dislocation. We routinely pool the sperm from 2 males to compensate for variability in sperm quality between males, generate sufficient quantities of straws for archiving and distribution and enable the optional quality control via an IVF of a single straw for each cryopreservation. Use of a single male is also possible but the number of straws and volume of media used needs to be reduced by 50%. The cauda epididymides and vasa deferentia were collected in DPBS (Sigma; D8537) and cleaned of fat and the testicular artery to avoid contaminating the sperm with blood. Next, the cauda epididymides were dried on a tissue (to avoid dilution of gCPA with DPBS) and transferred to a 240 µl drop of gCPA prewarmed on a 37˚C hot plate for at least 5 min. After making 6–7 cuts across the cauda epididymides with a pair of micro spring scissors the dish was placed on a 37˚C hot plate for 3 min and gently swirled every min for 20 sec to help the sperm disperse from the tissue. Following incubation, the sperm suspension was divided into 20 aliquots of 10 µl on a 10-cm culture dish lid while avoiding carryover of paraffin oil into the aliquots (the pipette tip was cleaned from the outside with a tissue to remove the oil before placing a 10 µl aliquot on the dish lid). Then each 10 µl drop was aspirated into a separate freezing straw followed by 2.3 cm air. Straws were sealed with metal balls (Minitüb GmbH; 13400/9970), placed in the liquid nitrogen gas phase for 10 min by using a dewar with a metal inlay and a triangular cassette (Cosmo Bio; KYD-S021) and then transferred to a liquid nitrogen tank for long-term storage. The integrity of each sperm sample after cryopreservation was evaluated by a validation IVF with oocytes from 3 superovulated females. Samples with fertilization rates of >20% were considered as successfully archived.

Before each IVF experiment a 35-mm culture dish with 90 µl c-TYH (or FERTIUP® PM) drop covered with paraffin oil was equilibrated overnight in a $CO_2$ incubator (5% $CO_2$, 37˚C, 95% humidity; Labotect C16). On the day of IVF, the required straw was removed from long-term storage in liquid nitrogen, placed in a small dewar of liquid nitrogen and then quickly transferred into a 37˚C water bath for 10 min. Afterwards, the straw was dried with a tissue and the metal ball-sealed end and the labeled end of the straw below the cotton plug were cut. Using a 1 ml syringe 10 µl of the sperm suspension was expelled into the center of a 90 µl c-TYH (or FERTIUP® PM) drop. If sperm cryopreserved via the Ostermeier *et al.* method were used the entire sperm suspension was placed onto the center of a 6-cm dish and 30 µl of that suspension added to a 90 µl c-TYH (or FERTIUP PM®) drop. If freshly harvested sperm were utilized 2 males of the same line were sacrificed and 2 x cauda epididymides (after removal of fat and blood) were transferred to separate dishes and into the oil next to the c-TYH (or FERTIUP PM®) drop. After nicking the tissue sperm were dragged with watchmaker forceps into the drop. Preincubation for 30 min (frozen sperm) or 60 min (freshly harvested sperm) in an atmosphere of mixed gas (5% $CO_2$, 5% $O_2$, 37˚C; Labotect C-Top) before the IVF procedure allowed capacitation of the sperm.

## *In vitro* fertilization

The CARD Set protocol was performed according to the manufacturer's instructions (Cosmo Bio; KYD-004-EX). RVF medium was used for the IVF procedure according to the Ostermeier *et al.* protocol. The IVF was performed as published [25]. Fertilization steps of all three

protocols were conducted in an atmosphere of mixed gas (5% $CO_2$, 5% $O_2$, 37˚C). Ideally IVF procedure should be carried out at 5% $O_2$ [25, 29–31] but atmospheric $O_2$ concentration works as well [27]. Oocytes were incubated in M16 immediately after fertilization.

The following description details the SEcuRe protocol developed in this study (Fig 2B). A 35-mm culture dish with a 90 μl (frozen sperm) or a 200 μl (freshly harvested sperm) drop of mRVF covered with oil for oocytes from a maximum of 3 (frozen sperm) or 5 (freshly harvested sperm) females was prepared and equilibrated for at least 20 min. Oocytes were obtained from 3-4-week-old (i.e., 12–14 g body weight) wild-type C57BL/6NRj or C57BL/6JRj females (Janvier Labs) superovulated with 5 IU of PMSG (pregnant mare's serum gonadotropin; Aviva Systems Biology; OPPA01037) followed after 48 hours by 5 IU of hCG (human chorionic gonadotropin; MSD Animal Health; Ovogest 300I.E.). Females were sacrificed by cervical dislocation 15 hours after the hCG injection. Oviducts were collected and after cleaning in DPBS transferred into the paraffin oil next to the 90 μl drop of mRVF. Oocyte clutches (oocytes with cumulus cells) were subsequently released into the oil by ripping the ampulla with forceps and dragged through the oil into the fertilization drop. Oocytes were incubated for 30–60 min before adding the sperm suspension. Afterwards, 10 μl (frozen) or 5μl (freshly harvested sperm of better quality which is usually based on a visual assessment) of the sperm suspension taken from the edge of the sperm capacitation drop was added to the oocyte clutches with the help of a 200 μl cell-saver tip (Biozym Scientific GmbH; 729055) and incubated for 3–5 hours. If the removal of cumulus cells assessed after 20 min of incubation was poor, indicating insufficient motility or concentration of sperm, additional 10 μl (frozen) or 5 μl (freshly harvested) of the sperm suspension was transferred to the fertilization medium.

## Embryo culture and transfer

After the IVF procedure embryos were washed through 10 drops of M16 to remove sperm excess and debris and incubated overnight in M16 in a $CO_2$ incubator (5% $CO_2$, 37˚C, 95% humidity) in groups of 15–50 embryos per drop (~30 μl drop of M16 covered with paraffin oil). The day after insemination, fertilization rates were determined and shown as a percentage of the total number of inseminated oocytes that developed to the 2-cell stage. To exclude counting parthenogenetic embryos from potentially unfertile males all IVFs with fertilization rate below 20% were removed from the analysis. 2-cell embryos were transferred unilaterally into the oviducts of pseudo-pregnant 0.5 dpc RjHan:NMRI females as described previously [17]. The birth rate was calculated from the number of pups born per embryos transferred to delivering recipients. The number of born pups was assessed one day after the expected delivery date. In some experiments, in order to assess the developmental capacity *in vitro*, 2-cell stage embryos were cultured in a $CO_2$ incubator (5% $CO_2$, 37˚C, 95% humidity) in M16 for 72 hours until the blastocyst stage. M16 medium for embryo culture was prepared according to the previously published method [17].

## Statistical analysis

Prism (GraphPad) was employed for the generation of graphs and calculation of statistical significance, standard deviation, arithmetic mean and median. Statistical significance was assessed using a one-way ANOVA (parametric) or a Kruskal–Wallis test (non-parametric) for comparison of 3 or more data sets or a two-tailed, unpaired Student's t-test for comparison of 2 data sets (parametric). Differences in the results were considered significant below a p-value of 0.05. The primary data used for analysis can be found in S2–S8 Tables.

## Results

We aimed to provide an economical and easily adaptable protocol for efficient IVF using frozen-thawed sperm of C57BL/6 mice. Our SEcuRe approach is based on the CARD method [27] as well as a subsequently established protocol from MRC Harwell available from the INFRAFRONTIER website [32]. This method includes all major improvements developed for inbred strains and has been shown to work well for C57BL/6 mice [23]. The three most important steps of our protocol involve: 1. Cryopreservation of sperm with L-glutamine supplemented CPA. The preparation of this solution is easily adaptable and cost effective as it contains only three simple components (raffinose pentahydrate, skim milk and L-glutamine) [27]. 2. Efficient capacitation with c-TYH containing MBCD. Although the composition of this medium has been described previously the complexity of the medium hinders swift adaptation. We have developed a simple protocol for preparation of c-TYH from concentrated stocks analogues to popular protocols for embryo culture medium preparation like KSOM [17] enabling easy adaptation. 3. Efficient IVF with supplementation of affordable commercially available RVF medium with GSH as well as elevation of the calcium concentration from default 2.04 mM to optimal 5.14 mM [19, 20, 33]. The composition of modified HTF, the fertilization medium used in the CARD protocol, has been published but in-house preparation of such complex media is challenging and requires quality control of each batch using zygotes [17]. Other IVF protocols therefore recommend the alternative use of commercially available RVF medium [20, 25, 33]. As the RVF medium does not contain GSH and has a calcium concentration of only 2.04 mM [33] the modifications described above are needed to prepare a fertilization medium according to state-of-the-art recommendations for IVF [19, 27, 33].

To validate our approach, we compared the *in vitro* fertilization rate of C57BL/6 oocytes with cryopreserved sperm from genetically engineered males in our SEcuRe method with either the CARD Set protocol or the method by Ostermeier *et al*. We analyzed the outcome of 137 independent experiments performed with the SEcuRe approach and found that out of a total of 16,873 inseminated oocytes 10,258 developed to 2-cell stage resulting in a median fertilization rate of 69.7% (Fig 3A). Strikingly, IVF with the CARD Set method utilizing commercially purchased FERTIUP® PM for sperm capacitation and CARD MEDIUM® for IVF led to the development of 1,551 2-cell embryos out of 2,582 inseminated oocytes achieving a virtually identical median fertilization rate of 68.1% in 9 independent experiments (Fig 3A). In contrast, during 79 independent IVFs conducted with the Ostermeier *et al*. protocol 3,029 out of 7,369 oocytes developed to the 2-cell stage yielding a significantly lower median fertilization rate in our hands (38.3%) than the SEcuRe ($p < 0.0001$) and the CARD Set approach ($p = 0.031$) (Fig 3A). Thus, we validated that our SEcuRe protocol performs equally well as the CARD Set method for C57BL/6 mice.

For convenient use in IVFs with frozen-thawed sperm the preincubation medium c-TYH can be commercially purchased as so called FERTIUP® PM. To provide a choice between convenience vs. cost-effectiveness we wanted to know whether c-TYH prepared in an economical fashion with our protocol can be exchanged for commercially available FERTIUP® PM. We therefore tested the fertilization capacity of sperm incubated with either in-house produced c-TYH medium or FERTIUP® PM side-by-side *in vitro*. Average fertilization rates attained 80.1% (235 out of 297 inseminated oocytes developed to 2-cell embryos) in IVFs with FERTIUP® PM and 81.4% (230 out of 283 inseminated oocytes reached the 2-cell stage) in IVFs with c-TYH, proving that both media perform equally effective (Fig 3B). In addition, we evaluated the developmental capacity of 2-cell stage embryos up to the blastocyst stage but did not observe any statistically significant difference ($p = 0.41$) between IVFs with FERTIUP® PM and c-TYH (Fig 3B). Thus, both sperm preincubation media can be used interchangeably in

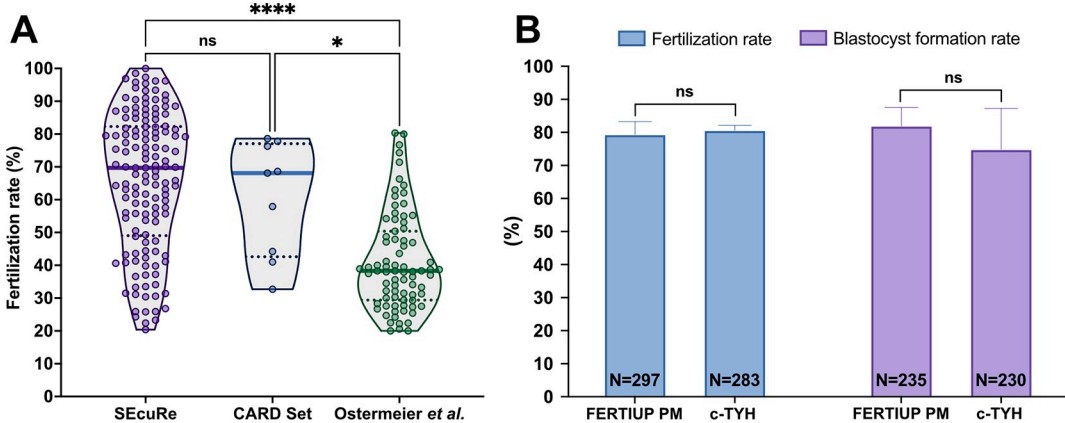

**Fig 3. Validation of the SEcuRe protocol.** (A) Comparison of the median fertilization rates obtained in the IVF procedures applying the SEcuRe, CARD Set and Ostermeier *et al.* protocols. Frozen-thawed sperm of C57BL/6 mutant males and oocytes of wild-type C57BL/6N or C57BL/6J females were used for the IVF procedures. Thick lines in the violin plots indicate median fertilization rates and dotted lines the first and the third quartile. Points represent individual experiments (IVF procedures). **** $p < 0.0001$; * $p = 0.031$; ns = non-significant ($p > 0.99$) (Kruskal–Wallis test). (B) Comparison of the average fertilization rates and blastocyst formation rates in the IVF procedures where either FERTIUP® PM or c-TYH was used as a sperm preincubation medium. The data was collected from three independent experiments. Frozen-thawed sperm of wild-type C57BL/6N males and oocytes of wild-type C57BL/6N females were used in these experiments. Data are shown as means ± standard deviation. The blastocyst formation rate shows the percentage of 2-cell embryos obtained in the IVF procedures that reached the blastocyst stage after 72 hours of *in vitro* culture. N indicates the total number of fertilized oocytes (fertilization rate) or 2-cell stage embryos (blastocyst formation rate). ns = non-significant ($p \geq 0.41$) (Student's t-test).

our SEcuRe approach providing researchers with an easy (FERTIUP® PM) or inexpensive (c-TYH) method for cryopreservation and IVF.

To confirm the developmental capacity of embryos generated via the SEcuRe protocol *in vivo* we compared the birth rates of 2-cell embryos produced with either IVF method after embryo transfer. As expected, we found no statistically significant difference between the mean birth rates upon the SEcuRe, CARD Set or Ostermeier *et al.* approach (31.4%, 32.8% and 29.1%, respectively; Table 1) validating, once again, the integrity of the SEcuRe approach.

In support of a broad applicability of our method we demonstrated that the SEcuRe IVF protocol, with slight modifications, can also be used with freshly harvested sperm or samples cryopreserved according to the Ostermeier *et al.* approach routinely used by The Jackson Laboratory [7] (S1A and S1B Fig). Furthermore, we tested our SEcuRe approach on sperm samples from a FVB/N mouse strain background and showed that the protocol can be successfully applied to other mouse strains as well (S1C Fig). Hence, we have provided evidence that the SEcuRe protocol can be used as a general method for cryopreservation of mice from different backgrounds, as well as, being applicable to rederivation using freshly harvested sperm, or sperm frozen using either of the most common protocols (i.e., CARD-based or Ostermeier *et al.*).

**Table 1. *In vivo* development of 2-cell stage embryos produced with the SEcuRe, CARD Set and Ostermeier *et al.* protocols.**

| | No. of experiments (n) | No. of 2-cell embryos transferred[a] | No. of recipient mice | No. of born pups | Birth rate (%)[b] |
|---|---|---|---|---|---|
| **SEcuRe protocol** | 33 | 2555 | 118 | 785 | 31.4%±9.6% |
| **CARD Set protocol** | 9 | 437 | 21 | 136 | 32.8%±11.0% |
| **Ostermeier *et al.* protocol** | 16 | 733 | 37 | 209 | 29.1%±10.0% |

[a] number of 2-cell embryos transferred into delivering recipient mice.

[b] no statistically significant differences between groups (one-way ANOVA; $p \geq 0.65$).

In conclusion, our SEcuRe approach provides researchers with a simple, inexpensive and flexible protocol for efficient cryopreservation and subsequent rederivation of genetically engineered C57BL/6 mice.

## Discussion

The new era of genome editing has accelerated the generation of GEMMs reinforcing the need for management of their numbers. Sperm cryopreservation represents an efficient and economic method for archiving and distribution of mice [2]. C57BL/6 is the most common background for GEMMs but sperm cryopreservation has long been challenging for this strain [4–7]. However, many improvements in the last decade led to a tremendous increase in the fertilization capacity of frozen-thawed C57BL/6 sperm. The CARD approach developed by Naomi Nakagata and Toru Takeo incorporates several major improvements for sperm cryopreservation and is therefore now widely used by many repositories [8, 23, 26, 27]. The required media are commercially available as FERTIUP® CPA, FERTIUP® PM and CARD MEDIUM® but purchase poses a constant burden on budgetary constraints. The in-house preparation of these media (gCPA, c-TYH and HTF modified with GSH and an elevated calcium concentration) is now possible [27] but the complex composition and the lack of detailed protocols for their preparation hinder broad applicability. Our goal was to offer an affordable and easily adaptable protocol for routine IVF using cryopreserved sperm based on the CARD method. We demonstrated that commercial RVF medium can be supplemented with GSH and calcium and performs equally well compared to CARD MEDIUM®. Similarly, our SEcuRe approach utilizes either FERTIUP® PM or in-house prepared c-TYH for sperm capacitation. As both c-TYH, prepared by our detailed protocol, and FERTIUP® PM performed equally well in terms of fertilization and developmental capacity we provide researchers with a choice between less expensive (c-TYH) or easily adaptable (FERTIUP® PM) variants of our SEcuRe method which will enable swift adaptation of this approach by any IVF laboratory.

In order to demonstrate integrity of our method we conducted side-by-side IVF experiments using the CARD Set (i.e., commercially available media for sperm preincubation and for fertilization). Subsequent comparison of fertilization rates *in vitro* and birth rates *in vivo* achieved with the SEcuRe and the CARD Set protocols clearly showed that our economic method yielded equal results. The robustness of the SEcuRe approach has been demonstrated in 137 individual IVF procedures which included a total of 16,873 inseminated oocytes out of which 10,258 developed to 2-cell stage. In addition, we compared the SEcuRe approach to a larger dataset generated by 79 individual C57BL/6 sperm cryopreservation and IVF procedures employing the method by Ostermeier *et al.* in our laboratory and revealed significantly lower fertilization rates for this mouse strain attained by the latter protocol. The methods by CARD and Ostermeier *et al.* represent the two main approaches for mouse cryopreservation and IVF [17] and our novel SEcuRe protocol is based on the same principles as the CARD method. These data are therefore the first comprehensive side-by-side comparison of the most popular cryopreservation and IVF approaches for C57BL/6 mice and indicate an advantage of the CARD-based protocols like ours.

A marked variability of fertilization rates upon IVF with cryopreserved sperm is visible in our experiments with each protocol tested. Apart from common variability of males even from the same line this likely arises from the fact that sperm used in our IVF procedures originated from males carrying different mutations potentially affecting their fertility. In fact, the influence of different genetic modifications on the quality of mouse sperm has been reported before even if the mutation has not been anticipated to affect reproductive performance [34–36]. This effect can even occur in a background specific fashion hindering immediate identification of

such mutations [37]. An additional factor increasing male to male variability is the age as older male mice tend to lose their fertilization capacity [34]. Results presented here are based on the retrospective studies where mainly 10- to 20-week-old males were used which is in accordance with common recommendations [24, 27]. Hence, the distinct variability in fertilization rates of cryopreserved sperm from the numerous GEMMs in our experiments is likely to closely recapitulate the actual variability in the daily routine in mouse IVF laboratories and is therefore expected to demonstrate a reliable median value.

To increase the applicability of the SEcuRe method, we have also demonstrated that our protocol can be easily adapted for use with freshly harvested sperm as well as sperm samples cryopreserved according to the Ostermeier *et al.* approach which is routinely used by JAX [7, 24, 25]. In this case sperm are cryopreserved in larger volumes of CPA leading to possibly lower sperm concentrations which could have hindered efficient fertilization rates when rederived with the CARD-based IVF protocols. Thus, we provide an easy method for rescue of cryopreserved, lower-concentrated sperm samples that yields comparable or higher fertilization rates than those of alternative methods involving elaborate centrifugation of sperm samples after thawing [33, 38]. These modifications are essentially adapted from the IVF protocol provided by MRC Harwell available from the INFRAFRONTIER website and are therefore shown to work in leading mouse repositories [32]. Hence, SEcuRe may be established as a universal protocol in a laboratory for any IVF independent of the sperm source.

A limitation of our approach is the use of a proprietary medium namely RVF. Although unlikely, the discontinuation of this product by the company would lead to inapplicability of our method. Nevertheless, RVF medium is similar to HTF medium which is available from many commercial vendors and its composition has been published [27]. HTF is therefore considered to be interchangeable with RVF medium during IVF [17, 25, 39]. Although not tested in this study, it is therefore highly likely that HTF can be utilized instead of RVF in our protocol as well.

Our method requires slightly more expertise than the Ostermeier *et al.* approach. The latter involves only one simple medium for the entire procedure presenting it as less error-prone for inexperienced users who may therefore benefit from applying the Ostermeier *et al.* method for sporadic use instead of ours. However, we demonstrate that the CARD-based cryopreservation/IVF protocols like ours show significantly higher fertilization rates using C57BL/6 sperm than those attained using the approach of Ostermeier *et al.*

Genome editing has led to a substantial surge in mouse model generation mostly on a C57BL/6 background. Subsequently the need for efficient cryopreservation of such models increased as well. As a consequence, application of the CARD-based methods may help minimize the number of mice utilized during IVF procedures in line with the principles of the 3Rs [40].

In summary, we have established a cost-effective and easily adaptable SEcuRe protocol that can be universally employed as one single protocol for IVFs as it works efficiently with frozen-thawed sperm generated by either of the common protocols (CARD or Ostermeier *et al.*) and can be easily modified to accommodate freshly harvested sperm.

## Supporting information

**S1 File. Step-by-step SEcuRe protocol, also available on protocols.io.**
(PDF)

**S1 Fig. Flexibility of the SEcuRe protocol.** Fertilization rates achieved after IVF using the SEcuRe IVF protocol with sperm samples from different sources and genetic backgrounds. The SEcuRe approach can be employed for IVF procedures utilizing: (A) freshly harvested

C57BL/6 sperm, (B) C57BL/6 sperm samples cryopreserved according to the Ostermeier *et al.* approach and (C) cryopreserved sperm samples from FVB/N background lines. Thick lines in the violin plots indicate median fertilization rates, dotted lines the first and the third quartile and points individual experiments (IVF procedures). For comparison, dashed lines demonstrate the median fertilization rate obtained with the SEcuRe protocol utilizing frozen-thawed C57BL/6 sperm in Fig 3A.
(PDF)

**S1 Table. The ARRIVE (Animal Research: Reporting of In Vivo Experiments) guidelines checklist.**
(PDF)

**S2 Table. Primary *in vitro* data–SEcuRe protocol.**
(PDF)

**S3 Table. Primary *in vitro* data–CARD Set protocol.**
(PDF)

**S4 Table. Primary *in vitro* data–Ostermeier *et al.* protocol.**
(PDF)

**S5 Table. Primary data–FERTIUP$^®$ PM–c-TYH comparison.**
(PDF)

**S6 Table. Primary *in vivo* data–SEcuRe protocol.**
(PDF)

**S7 Table. Primary *in vivo* data–CARD Set protocol.**
(PDF)

**S8 Table. Primary *in vivo* data–Ostermeier *et al.* protocol.**
(PDF)

**S9 Table. Number of mice used during IVF procedures and embryo transfers.**
(PDF)

## Acknowledgments

We thank Sonja Assenmacher, Anni Feldmann, Marco Schneider, Patrick Jankowski, Kerstin Weisheit, Theresa Tschanz and Stefanie Wasserburger for excellent technical support. We are grateful to Sonja Assenmacher for her valuable comments during the preparation of the manuscript and the SEcuRe protocol. Figures were partially created with BioRender.com.

## Author Contributions

**Conceptualization:** Magdalena Wigger, Simon E. Tröder, Branko Zevnik.

**Formal analysis:** Magdalena Wigger.

**Funding acquisition:** Branko Zevnik.

**Investigation:** Magdalena Wigger, Simon E. Tröder.

**Methodology:** Magdalena Wigger, Simon E. Tröder.

**Supervision:** Simon E. Tröder, Branko Zevnik.

**Validation:** Magdalena Wigger.

**Visualization:** Magdalena Wigger.

**Writing – original draft:** Magdalena Wigger, Simon E. Tröder, Branko Zevnik.

**Writing – review & editing:** Magdalena Wigger, Simon E. Tröder, Branko Zevnik.

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
