## [Decision Letter · Decision Letter 0]

3 Sep 2021

PONE-D-21-23832

A simple and economic protocol for efficient in vitro fertilization using cryopreserved mouse sperm

PLOS ONE

Dear Dr. Zevnik,

Thank you for submitting your manuscript to PLOS ONE. After careful consideration, we feel that it has merit but does not fully meet PLOS ONE’s publication criteria as it currently stands. Therefore, we invite you to submit a revised version of the manuscript that addresses the points raised during the review process.

This is an intersting and valuable manuscripts and all three referees recommend its publication. The authors need to make the recommended minor changes and correction to improve the paper before a final decision is made.

We look forward to receiving your revised manuscript.

Kind regards,

Stefan Schlatt

Academic Editor

PLOS ONE

Journal Requirements:

Reviewers' comments:

Reviewer's Responses to Questions

**Comments to the Author**

1. Does the manuscript report a protocol which is of utility to the research community and adds value to the published literature?

Reviewer #1: Yes

Reviewer #2: Yes

Reviewer #3: Yes

2. Has the protocol been described in sufficient detail?

Descriptions of methods and reagents contained in the step-by-step protocol should be reported in sufficient detail for another researcher to reproduce all experiments and analyses. The protocol should describe the appropriate controls, sample sizes and replication needed to ensure that the data are robust and reproducible.

Reviewer #1: Partly

Reviewer #2: Yes

Reviewer #3: Yes

3. Does the protocol describe a validated method?

Reviewer #1: Yes

Reviewer #2: Yes

Reviewer #3: Yes

4. If the manuscript contains new data, have the authors made this data fully available?

Reviewer #1: Yes

Reviewer #2: Yes

Reviewer #3: Yes

**5. Is the article presented in an intelligible fashion and written in standard English?**

Reviewer #1: Yes

Reviewer #2: Yes

Reviewer #3: Yes

6. Review Comments to the Author

Reviewer #1: This manuscript contributes an improved protocol to cryopreserve mouse spermatozoa for subsequent use in the rederivation of genetically engineered mouse strains, which are often based on the C57Bl/6 background. The manuscript builds on a retrospective study. The manuscript is clearly written and the protocol provides sufficient detail to enable repetition of the procedure, which has been validated (in vitro and in vivo embryo development).

My remarks are technical in nature.

The Authors relied on M16 as embryo culture medium. This is not wrong, of course, but this choice of medium is outdated, considering the progress made in the field since the introduction of KSOM. Using M16 means: lower blastocyst rates and lower blastocyst fitness, compared to KSOM(aa), due to the high (M16) glucose concentration if nothing else. In my opinion, the Authors gained miles during the cryopreservation / thawing / IVF, only to lose those extra miles to a suboptimal medium. For future work I advise the Authors to consider switching to a more modern medium than M16.

I was not able to find the sperm concentration used in IVF. The Authors mention sperm concentration in passing at lines 278, 447 and 449, but I am left to wonder what the approximate values were. When the fertilization rate is below 20%, is this because of the sperm quality or because of its concentration? In my own experience (although I use a different protocol e.g. Whittingham’s medium with 3% BSA, and I select sperm by swim-up) I need 2 million sperm / ml for reliable fertilization of cumulus-free mouse oocytes; in my hands IVF works inconsistently or not all all below 1 million sperm / mL. I think the Authors should provide some information about the range of sperm concentrations they experienced in their IVF experiments.

The B6 females used for superovulation are very young at the age of 3-4 weeks. They weigh only 12-14 grams. I use F1 of 8 weeks and 25g, so I must admit I can’t compare to B6, but I’d think that females can barely be weaned at the age of 3-4 weeks, and they are just slightly older than puberty. If my animal caretakers saw me using such young females for experiments, they’d approach me and question my choice. My surprise with the 3-4 week-old and the 12-14 grams heavy females has also to do with the oocyte quality. Essentially, the Authors are using oocytes from the first wave of ovulation. How good is the quality?

The birth rate (Table 1) is about 30% after surgical transfer of 2-cell embryos to oviduct. Does this include all females, or only those that got pregnant? I seem to understand from the supplementary material that all recipients were considered. Please elaborate on why the birth rate was not closer to 100%, apart from the technical error. Of course, surgical ET is not a perfect procedure in general, because it is invasive; it is expected that some surgical transfers will fail in part due to failed retention of the embryos in the oviduct. What else could contribute to the incomplete birth rates? Embryo quality? Please also indicate whether the birth rate includes all pups or only those which were really fostered (as opposed to being cannibalized by the mother shortly after birth).

The statistical analysis of the data relied on t test and ANOVA, that is, parametric tests. Looking at the violin plots of Figure 3, I am not sure if the frequencies are normally distributed. It might be safer to use non-parametric tests.

I hope the Authors find my comments useful.

Reviewer #2: General comments:

The continual development and dissemination of simplified methods for sperm cryopreservation and IVF recovery are essential if the community is going fully embrace alternatives to maintaining live colonies and/or exchanging live mice between laboratories. For this reason, I think the article should be published.

I would urge the author to be careful in the language they use when drawing comparisons with the CARD/SECURE and Ostermeier methods. We have also found the Ostermeier method does not perform as well in our laboratory as the CARD based methods but I know of other labs, including the Jax where the Ostermeier method performs very well. The differences between Ostermeier and CARD/SEcure may to some extent simply reflect the different ways we interpret the protocols.

I notice that the authors have used ketamine and xylazine chloride for anaesthesia. Whilst these injectable agents give good anaesthesia and their use is well understood, I would encourage the authors to investigate the use of gaseous anaesthetics in the future. For example, isoflurane is well tolerated by mice, induces rapid anaesthesia and the mice recover very quickly once they have been removed from the anaesthetic machine.

Specific comments:

Line 79 – Change the sentence to ‘ The comparison of the IVF medium used in conjunction with frozen-thawed ….’

Line 89 - change the sentence to ‘Two protocols for sperm cryopreservation and IVF incorporating these improvements are commonly used ….’

Line 100 – delete reference to the Medical Research Council

Line 107 – change sentence to ‘….method for IVF using cryopreserved C57BL/6 sperm for routine ….’

Line 122 – change sentence to ‘….(when FERTIUP PM is utilized) and more economical (c-TYH) approach to IVF recoveries.’

Line 127 – change the word ‘with’ to ‘using’

Line 180 – change the word ‘applied’ to ‘used’

Line 201 – change the sentence to ‘….explanation of the sperm freezing/thawing and capacitation protocol employed in the Ostermeier et al method has been described previously [24, 25]’

Line 205 - change sentence to ‘…4.0cm at the open end…’

Line 224 – change sentence to ‘….male is also possible but the number of straws and volume of media used needs to be reduced by 50%.’

Line 235 – explain what is meant by triangular cassette or show a picture

Line 242 – change the wording to …was removed from long term storage in liquid nitrogen, placed in a small dewar of liquid nitrogen…’

Line 247 – change ‘were’ to ‘was’

Line 250 change wording to ‘…2 x cauda epididymides (after removal of fat and blood) were transferred to separate dishes and into the…’

Line 265 – change wording to ‘…for oocytes from a maximum of 3 (frozen sperm)….’

Line 275 – change ‘selected’ to ‘which is usually’

Line 276 – change sentence to ‘..sperm suspension taken from the edge of the sperm capacitation drop was added…’

Line 290 – unilateral should read ‘unilateral’

Line 297 – change sentence to ‘Prism (Graph) was employed for the generation of graphs and calculation of statistical significance, …’

Line 299 – change >2 to ‘2 or more’

Line 321 – change to ‘As the RVF medium does not’

Line 322 – change to …[33] the modifications described above are needed to prepare..’

Line 325 – change to ‘… compared the in vitro fertilization rate of C57BL/6 oocytes with cryopreserved sperm from genetically engineered males in our SEcuRe method with either the …’

Line 337 – change to ‘…equally well as the CARD…’

Line 356 – change to ‘…a choice between convenience vs cost-effectiveness …’

Line 358 – remove ‘for full flexibility’

Line 364 – change to ‘…stage embryos up to the blastocyst stage but did not …’

Line 383 – change to ‘…samples from mouse strains on an FVB/N background and showed that ..’

Line 385 change to ‘… Hence, we have provided evidence that the ‘

Line 386 change universal to ‘general’

Line 386 – change to ‘…backgrounds, as well as, being applicable to rederivation using freshly harvested sperm, or sperm frozen using either of the …’

Line 400 – change ‘all’ to ‘several’

Line 401 – remove EMMA

Line 409 – change to ‘…CARD MEDIUM. Similarly, our SEcuRe approach utilizes either …’

Line 410 – remove the word ‘reliably’

Line 428 – change to ‘…demonstrate the same advantages of the CARD-based protocols in our laboratory.’

Line 458 change to ‘HTF is therefore considered to be interchangeable …’

Line 462 – remove ‘In addition’

Line 467 start a new paragraph at ‘Genome editing….’

Line 474 change to ‘and can be easily modified to …’

Reviewer #3: This is a very valuable paper for the lab animal science community. The authors present a cost effective and efficient protocol as an alternative to the two mostly wide used protocols (Ostermeier et al. and CARD) for sperm cryopreservation and IVF which they name SEcuRe. Additionally, they present within the protocol two different approaches that are either less complicated or more economical depending on the financial capacity or stuff experience. All necessary steps and procedures are well described in all details for reproducibility. The authors present an impressive amount of empirical data when comparing the different methods. Although the amount of the data sets of the different protocols are not identical, this does not present a problem as the large data set of the SEcuRe protocol proves its robust effectiveness comparable or even better to the known standards. As they successfully tested it not only in the most often used B6 background lines but also in FVB/N lines it appears to be a universal method for different background lines.

The publication has the high potential to enhance further the progress in mouse artificial reproductive technologies if the presented protocol with its flexible approaches will be tested and implemented by other labs.

7. PLOS authors have the option to publish the peer review history of their article (what does this mean?). If published, this will include your full peer review and any attached files.

Reviewer #1: **Yes: **Michele Boiani

Reviewer #2: No

Reviewer #3: No

---

## [Author Response · Author response to Decision Letter 0]

10 Sep 2021

We would like to thank all reviewers for their valuable time and their careful and thorough reviews. Their positive comments regarding the quality of our manuscript and data are motivating and we strongly believe that we have now addressed all of their concerns by rephrasing the manuscript according to their suggestions. Please find our ‘point-by-point’ responses to the reviewers’ comments below.

Reviewer #1: 

The Authors relied on M16 as embryo culture medium. This is not wrong, of course, but this choice of medium is outdated, considering the progress made in the field since the introduction of KSOM. Using M16 means: lower blastocyst rates and lower blastocyst fitness, compared to KSOM(aa), due to the high (M16) glucose concentration if nothing else. In my opinion, the Authors gained miles during the cryopreservation / thawing / IVF, only to lose those extra miles to a suboptimal medium. For future work I advise the Authors to consider switching to a more modern medium than M16.

Response: We thank the reviewer for this suggestion regarding future experiments. We entirely agree that KSOM(aa) is well documented to be advantageous over M16 for culture of mouse pre-implantation embryos and that many laboratories have therefore changed to KSOM(aa). Accordingly, we point out that KSOM(aa) may be used instead of M16 in our protocol on protocols.io. However, numerous comparisons of KSOM(aa) (either purchased or produced in-house) and M16 side-by-side with C57BL/6N embryos in our lab consistently showed significantly higher developmental capacity of the embryos cultured in M16. In fact, we routinely reach blastocyst rates as high as 80% as seen in Fig 3B. Although we do not have any explanation for this discrepancy with published data yet, the consistently superior results led us to continue embryo culture using M16 in our lab. Nevertheless, we of course agree that the majority of labs are successfully working with KSOM(aa) and have now modified the protocol on protocols.io to better reflect the recommendation of KSOM(aa) by the current literature. 

I was not able to find the sperm concentration used in IVF. The Authors mention sperm concentration in passing at lines 278, 447 and 449, but I am left to wonder what the approximate values were. When the fertilization rate is below 20%, is this because of the sperm quality or because of its concentration? In my own experience (although I use a different protocol e.g. Whittingham’s medium with 3% BSA, and I select sperm by swim-up) I need 2 million sperm / ml for reliable fertilization of cumulus-free mouse oocytes; in my hands IVF works inconsistently or not all all below 1 million sperm / mL. I think the Authors should provide some information about the range of sperm concentrations they experienced in their IVF experiments.

Response: We apologize for the confusion and would like to thank the reviewer very much for pointing out a misleading description in our manuscript. The CARD-based cryopreservation and IVF protocols do not include the determination of the sperm concentration directly but assess the motility and concentration of the sperm only indirectly by observing the disintegration of the cumulus complex as a known indicator for successful fertilization [17, 27, 32]. We have therefore not documented values for the sperm concentration. We have now rephrased the corresponding sentences in the revised version of the manuscript (line 278, 447 and 449; original manuscript) and on protocols.io to explain this procedure clearer. We agree that it will be interesting to determine whether reduced motility or low concentration of the sperm is responsible for the low fertilization capacity of certain males during IVF procedures. However, this was not the goal of our current study since we aimed to provide an easily adaptable and cost-efficient alternative to the CARD IVF and cryopreservation protocol. 

The B6 females used for superovulation are very young at the age of 3-4 weeks. They weigh only 12-14 grams. I use F1 of 8 weeks and 25g, so I must admit I can’t compare to B6, but I’d think that females can barely be weaned at the age of 3-4 weeks, and they are just slightly older than puberty. If my animal caretakers saw me using such young females for experiments, they’d approach me and question my choice. My surprise with the 3-4 week-old and the 12-14 grams heavy females has also to do with the oocyte quality. Essentially, the Authors are using oocytes from the first wave of ovulation. How good is the quality?

Response: The timing for superovulation is highly strain dependent and indeed may even impact the quality of the harvested embryos. As C57BL/6 is one of the most common background for genetically engineered mice many studies have already established the optimal superovulation parameters for this strain. In accordance with these studies, we use females of 3-4 weeks of age (corresponding to 12-14g) for superovulation for optimal oocyte quality. In support we would like to refer to two highly cited publication (PMID: 16271754 and PMID: 21838974) as well as “The Mouse Manual” - Manipulating the Mouse Embryo [17] recommending 3-4-week-old C57BL/6 females as optimal for superovulation. We are therefore confident that the oocytes in our current study are of high quality.

The birth rate (Table 1) is about 30% after surgical transfer of 2-cell embryos to oviduct. Does this include all females, or only those that got pregnant? I seem to understand from the supplementary material that all recipients were considered. Please elaborate on why the birth rate was not closer to 100%, apart from the technical error. Of course, surgical ET is not a perfect procedure in general, because it is invasive; it is expected that some surgical transfers will fail in part due to failed retention of the embryos in the oviduct. What else could contribute to the incomplete birth rates? Embryo quality? Please also indicate whether the birth rate includes all pups or only those which were really fostered (as opposed to being cannibalized by the mother shortly after birth).

Response: We would like to thank the reviewer for bringing a potentially insufficient explanation of this analysis to our attention. As indicated in the footnote (a) of Table 1 of our original manuscript the live birth rates are calculated from delivering females. We have now tried to explain this clearer with an additional explanation in the Materials and Methods section (line 291; original manuscript) and Table 1 of the revised version of the manuscript. We have now also stated more precisely that the birth rates include all born pups and not only those fostered (line 292; original manuscript) as pointed out by the reviewer at the end of this paragraph. 

We agree that surgical embryo transfer is indeed never perfect. For clarification we would like to point out, that “birth rate” refers to pups born/transferred embryos, and not to females delivering. While the latter value is significantly higher, we believe that 30% pups born from transferred embryos are commonly expected after surgical transfer of 2-cell stage embryos from genetically engineered C57BL/6 mice. We would like to refer to four large-scale studies utilizing such embryos generated by either IVF ([33] and PMID: 25080098) or even natural mating (PMID: 30866727 and PMID: 29554999) which show ranges of birth rates from about 30 to 40%. As we are well in range of these values, we feel confident that the embryos in our study are of high quality. 

The statistical analysis of the data relied on t test and ANOVA, that is, parametric tests. Looking at the violin plots of Figure 3, I am not sure if the frequencies are normally distributed. It might be safer to use non-parametric tests.

Response: We would like to thank the reviewer for raising this important point. We have now applied a non-parametric Kruskal-Wallis test instead of a one-way ANOVA for the presumably non-Gaussian distributed data of the violin plots in Fig. 3. As the statistical significance and subsequently p-value categories did not change we did not update the figures’ appearance but only updated the p-values in the text. We also checked and enhanced the explanation of our statistical analysis for Table 1. We did not change the hypothesis test (one-way ANOVA) as the distribution of values was Gaussian but now indicated the precise p-values. 

I hope the Authors find my comments useful. 

Response: We very much do and would like to thank Dr. Boiani for his thorough review which helped to improve our manuscript. 

Reviewer #2:

I would urge the author to be careful in the language they use when drawing comparisons with the CARD/SECURE and Ostermeier methods. We have also found the Ostermeier method does not perform as well in our laboratory as the CARD based methods but I know of other labs, including the Jax where the Ostermeier method performs very well. The differences between Ostermeier and CARD/SEcure may to some extent simply reflect the different ways we interpret the protocols.

Response: We appreciate the reviewers’ comment to alleviate our wording in this regard. We have now rephrased any sentence with comparison of CARD-based and Ostermeier-based IVF methods accordingly throughout the entire manuscript (i.e., line 119, 128, 334-338; 423 and 428; original manuscript)

I notice that the authors have used ketamine and xylazine chloride for anaesthesia. Whilst these injectable agents give good anaesthesia and their use is well understood, I would encourage the authors to investigate the use of gaseous anaesthetics in the future. For example, isoflurane is well tolerated by mice, induces rapid anaesthesia and the mice recover very quickly once they have been removed from the anaesthetic machine.

Response: We agree that gaseous anaesthetics like isoflurane may be advantageous over conventional ketamine and xylazine and would like to thank the reviewer for pointing this out. In fact, we are currently in the process of changing to isoflurane for anesthesia. As we used ketamine and xylazine throughout the current study, we did indicate this in the Materials and Methods section. Nevertheless, following the reviewers’ suggestion we have now added a recommendation for gaseous anaesthetics like isoflurane to our protocol on protocols.io to encourage its use over ketamine and xylazine.

Specific comments:

Line 79 – Change the sentence to ‘ The comparison of the IVF medium used in conjunction with frozen-thawed ….’

Line 89 - change the sentence to ‘Two protocols for sperm cryopreservation and IVF incorporating these improvements are commonly used ….’

Line 100 – delete reference to the Medical Research Council

Line 107 – change sentence to ‘….method for IVF using cryopreserved C57BL/6 sperm for routine ….’

Line 122 – change sentence to ‘….(when FERTIUP PM is utilized) and more economical (c-TYH) approach to IVF recoveries.’

Line 127 – change the word ‘with’ to ‘using’

Line 180 – change the word ‘applied’ to ‘used’

Line 201 – change the sentence to ‘….explanation of the sperm freezing/thawing and capacitation protocol employed in the Ostermeier et al method has been described previously [24, 25]’

Line 205 - change sentence to ‘…4.0cm at the open end…’

Line 224 – change sentence to ‘….male is also possible but the number of straws and volume of media used needs to be reduced by 50%.’

Line 235 – explain what is meant by triangular cassette or show a picture

Line 242 – change the wording to …was removed from long term storage in liquid nitrogen, placed in a small dewar of liquid nitrogen…’

Line 247 – change ‘were’ to ‘was’

Line 250 change wording to ‘…2 x cauda epididymides (after removal of fat and blood) were transferred to separate dishes and into the…’

Line 265 – change wording to ‘…for oocytes from a maximum of 3 (frozen sperm)….’

Line 275 – change ‘selected’ to ‘which is usually’

Line 276 – change sentence to ‘..sperm suspension taken from the edge of the sperm capacitation drop was added…’

Line 290 – unilateral should read ‘unilateral’

Line 297 – change sentence to ‘Prism (Graph) was employed for the generation of graphs and calculation of statistical significance, …’

Line 299 – change >2 to ‘2 or more’

Line 321 – change to ‘As the RVF medium does not’

Line 322 – change to …[33] the modifications described above are needed to prepare..’

Line 325 – change to ‘… compared the in vitro fertilization rate of C57BL/6 oocytes with cryopreserved sperm from genetically engineered males in our SEcuRe method with either the …’

Line 337 – change to ‘…equally well as the CARD…’

Line 356 – change to ‘…a choice between convenience vs cost-effectiveness …’

Line 358 – remove ‘for full flexibility’

Line 364 – change to ‘…stage embryos up to the blastocyst stage but did not …’

Line 383 – change to ‘…samples from mouse strains on an FVB/N background and showed that ..’

Line 385 change to ‘… Hence, we have provided evidence that the ‘

Line 386 change universal to ‘general’

Line 386 – change to ‘…backgrounds, as well as, being applicable to rederivation using freshly harvested sperm, or sperm frozen using either of the …’

Line 400 – change ‘all’ to ‘several’

Line 401 – remove EMMA

Line 409 – change to ‘…CARD MEDIUM. Similarly, our SEcuRe approach utilizes either …’

Line 410 – remove the word ‘reliably’

Line 428 – change to ‘…demonstrate the same advantages of the CARD-based protocols in our laboratory.’

Line 458 change to ‘HTF is therefore considered to be interchangeable …’

Line 462 – remove ‘In addition’

Line 467 start a new paragraph at ‘Genome editing….’

Line 474 change to ‘and can be easily modified to …’

Response: We would very much like to thank the reviewer for these specific comments which we have fully implemented in the revised version of the manuscript and the protocol on protocols.io. We took the liberty of the following exceptions: in line 79 we assume the reviewer meant to suggest “composition” instead of “comparison” and in line 290 “unilaterally” instead of “unilateral” and adapted this accordingly. In line 247 we left “were” for consistency reasons. In line 299 we changed the “>2” to “3 or more” instead of the suggested “2 or more” to preserve the correct meaning. In line 383 we modified the reviewers’ suggestion slightly to “samples from a FVB/N mouse strain background” to properly reflect mutant FVB/N lines. We very much appreciate the time the reviewer spent to improve our manuscript!

Reviewer #3: 

This is a very valuable paper for the lab animal science community. The authors present a cost effective and efficient protocol as an alternative to the two mostly wide used protocols (Ostermeier et al. and CARD) for sperm cryopreservation and IVF which they name SEcuRe. Additionally, they present within the protocol two different approaches that are either less complicated or more economical depending on the financial capacity or stuff experience. All necessary steps and procedures are well described in all details for reproducibility. The authors present an impressive amount of empirical data when comparing the different methods. Although the amount of the data sets of the different protocols are not identical, this does not present a problem as the large data set of the SEcuRe protocol proves its robust effectiveness comparable or even better to the known standards. As they successfully tested it not only in the most often used B6 background lines but also in FVB/N lines it appears to be a universal method for different background lines.

The publication has the high potential to enhance further the progress in mouse artificial reproductive technologies if the presented protocol with its flexible approaches will be tested and implemented by other labs.

Response: We thank this reviewer for his/her very positive assessment of our work!

---

## [Decision Letter · Decision Letter 1]

15 Oct 2021

A simple and economic protocol for efficient in vitro fertilization using cryopreserved mouse sperm

PONE-D-21-23832R1

Dear Dr. Zevnik,

We’re pleased to inform you that your manuscript has been judged scientifically suitable for publication and will be formally accepted for publication once it meets all outstanding technical requirements.

Kind regards,

Stefan Schlatt

Academic Editor

PLOS ONE

Additional Editor Comments (optional):

Reviewers' comments:

Reviewer's Responses to Questions

**Comments to the Author**

1. Does the manuscript report a protocol which is of utility to the research community and adds value to the published literature?

Reviewer #1: Yes

2. Has the protocol been described in sufficient detail?

Descriptions of methods and reagents contained in the step-by-step protocol should be reported in sufficient detail for another researcher to reproduce all experiments and analyses. The protocol should describe the appropriate controls, sample sizes and replication needed to ensure that the data are robust and reproducible.

Reviewer #1: Yes

3. Does the protocol describe a validated method?

Reviewer #1: Yes

4. If the manuscript contains new data, have the authors made this data fully available?

Reviewer #1: Yes

**5. Is the article presented in an intelligible fashion and written in standard English?**

Reviewer #1: Yes

6. Review Comments to the Author

Reviewer #1: My previous critiques have been addressed to my full satisfaction. I approve the revised version of the manuscript.

7. PLOS authors have the option to publish the peer review history of their article (what does this mean?). If published, this will include your full peer review and any attached files.

Reviewer #1: **Yes: **Michele Boiani

---

## [Editor Report · Acceptance letter]

20 Oct 2021

PONE-D-21-23832R1 

A simple and economic protocol for efficient *in vitro* fertilization using cryopreserved mouse sperm 

Dear Dr. Zevnik:

I'm pleased to inform you that your manuscript has been deemed suitable for publication in PLOS ONE. Congratulations! Your manuscript is now with our production department. 

Kind regards, 

on behalf of

Dr. Stefan Schlatt 

Academic Editor

PLOS ONE